

# Determination of Primary combustion source organic carbon-to-elemental carbon (OC/EC) ratio using ambient OC and EC measurements: Secondary OC-EC correlation minimization method

**Cheng Wu[1], Jian Zhen Yu[1, 2, 3]**

[1] Division of Environment, Hong Kong University of Science and Technology, Clear Water Bay, Hong Kong, China

[2] Atmospheric Research Centre, Fok Ying Tung Graduate School, Hong Kong University of Science and Technology, Nansha, China

[3] Department of Chemistry, Hong Kong University of Science and Technology, Clear Water Bay, Hong Kong, China

Correspondence to: Jian Zhen Yu (jian.yu@ust.hk)

**Abstract**

Elemental carbon (EC), due to its exclusive origin in primary combustion sources, has been widely used as a tracer to track the portion of co-emitted primary organic carbon (OC) and, by extension, to estimate secondary OC (SOC) from ambient observations of EC and OC. Key to this EC tracer method is to determine an appropriate OC/EC ratio that represents primary combustion emission sources (i.e., $(OC/EC)_{pri}$) at the observation site. The conventional approaches include regressing OC against EC within a fixed percentile of the lowest (OC/EC) ratio data (usually 5-20%) or relying on a subset of sampling days with low photochemical activity and dominated by local emissions. The drawback of these approaches is rooted in its empirical nature, i.e., a lack of clear quantitative criteria in the selection of data subsets for the $(OC/EC)_{pri}$ determination. We examine here a method that derives $(OC/EC)_{pri}$ through calculating a hypothetical set of $(OC/EC)_{pri}$ and SOC followed by seeking the minimum of the coefficient of correlation ($R^2$) between SOC and EC. The hypothetical $(OC/EC)_{pri}$ that generates the minimum $R^2(SOC,EC)$ then represents the actual $(OC/EC)_{pri}$ ratio if variations of EC and SOC are independent. This Minimum R Squared (MRS) method





has a clear quantitative criterion for the $(OC/EC)_{pri}$ calculation. The general concept embodied in the MRS method was initially proposed by Miller et al (2005), but has not been
evaluated for accuracy or utility since its debut. This work uses numerically simulated data to evaluate the accuracy of SOC estimation by the MRS method and to compare with two
commonly used methods: minimum OC/EC ($OC/EC_{min}$) and OC/EC percentile ($OC/EC_{10\%}$). Log-normally distributed EC and OC concentrations with known proportion of SOC are
numerically produced through a Mersenne twister pseudorandom number generator. Three scenarios are considered, including a single primary source, two independent primary sources,
and two correlated primary sources. Among the three SOC estimation methods, the MRS method consistently yields the most accurate SOC estimation. Unbiased SOC estimation by
$OC/EC_{min}$ and $OC/EC_{10\%}$ only occur when the left tail of OC/EC distribution is aligned with the peak of the $(OC/EC)_{pri}$ distribution, which is fortuitous rather than norm. In contrast,
MRS provides an unbiased SOC estimation since it is insensitive to the position of $(OC/EC)_{pri}$ relative to the (OC/EC) distribution. Sensitivity tests of OC and EC measurement
uncertainty on SOC estimation demonstrate the superior accuracy of MRS over the other two approaches.



## 1 Introduction

Organic carbon (OC) and elemental carbon (EC) are among the major components of fine
particular matter ($PM_{2.5}$) (Malm et al., 2004). EC is a product of carbon fuel-based

combustion processes and is exclusively associated with primary emissions whereas OC can
be from both direct emissions and be formed through secondary pathways. Differentiation

between primary organic carbon (POC) and secondary organic carbon (SOC) is indispensable
for probing atmospheric aging processes of organic aerosols and formulating effective

emission control policies. However, direct SOC measurement is not yet feasible, as there
lacks knowledge of its chemical composition at the molecular level. Due to its exclusive

origin in primary combustion sources, EC was first proposed by Turpin and Huntzicker (1991)
to serve as the tracer to track POC from primary combustion sources and, by extension, to

estimate SOC as SOC is simply the difference between OC and POC. This EC tracer method
only requires measurements of OC and EC. Due to its simplicity, the EC tracer method has

been widely adopted in studies reporting ambient OC and EC measurements (e.g., Castro et
al., 1999; Cao et al., 2004; Yu et al., 2004). If OC and EC concentrations are available and

primary OC from non-combustion sources ($OC_{non-comb}$) is negligible, SOC can be estimated
using EC as the tracer for combustion source POC (Turpin and Huntzicker, 1995):

$$POC = (OC/EC)_{pri} \times EC \qquad (1)$$

$$SOC = OC_{total} - (OC/EC)_{pri} \times EC \qquad (2)$$

where $(OC/EC)_{pri}$ is the OC/EC ratio in freshly emitted combustion aerosols, and $OC_{total}$ and
EC are available from ambient measurements.

The key step in the EC tracer method is to determine an appropriate OC/EC ratio that
represents primary combustion emission sources (i.e., $(OC/EC)_{pri}$) at the observation site.

Various approaches in deriving $(OC/EC)_{pri}$ reported in the literature are either based on
emission inventory (Gray et al., 1986) or ambient observation data. Using ambient

observation data, three approaches are the most common: 1) regressing measured OC vs. EC
data from times of low photochemical activity and dominated by local emissions; 2)

regressing measured OC vs. EC data on a fixed percentile of the lowest OC/EC ratio (usually
5-20%) data to represent samples dominated by primary emissions (Lim and Turpin, 2002;

Lin et al., 2009); and 3) simply taking the minimum OC/EC ratio during the study period to
approximate $(OC/EC)_{pri}$ (Castro et al., 1999). These approaches have the drawback in that

there is not a clear quantitative criterion in the data selection for the $(OC/EC)_{pri}$ determination.





Millet et al. (2005) was the first to propose an algorithm that explores the inherent
independency between pollutants from primary emissions (e.g., EC) and products of
secondary formation processes (e.g., SOC) to derive the primary ratios (e.g., $(OC/EC)_{pri}$) for
species with multiple source types. More specifically, for the determination of $(OC/EC)_{pri}$, the
assumed $(OC/EC)_{pri}$ value is varied continuously. At each hypothetical $(OC/EC)_{pri}$, SOC is
calculated for the data set and a correlation coefficient value ($R^2$) of EC vs. SOC (i.e.,
$R^2(EC,SOC)$) is generated. The series of $R^2(EC,SOC)$ values are then plotted against the
assumed $(OC/EC)_{pri}$ values. If variations of EC and SOC are independent, the assumed
$(OC/EC)_{pri}$ corresponding to the minimum $R^2(EC,SOC)$ would then represent the actual
$(OC/EC)_{pri}$ ratio. Such an approach obviates the need for an arbitrary selection criterion, as
the algorithm seeks the minimum point, which is unique to the dataset. However, this method
has largely been overlooked, with only one study reporting its use (Hu et al., 2012) since its
debut, which may be a result of a lack of evaluation of its method performance. Hereafter for
the convenience of discussion, we call this method the minimum $R$ squared (MRS) method,
with a conceptual illustration of the MRS method shown in Figure 1. A computer program
written in Igor Pro (WaveMetrics, Inc. Lake Oswego, OR, USA) is developed to feasible
MRS calculation and it is available from https://sites.google.com/site/wuchengust.

With ambient OC and EC samples, the accuracy of estimated SOC by different $(OC/EC)_{pri}$
methods is difficult to evaluate due to the lack of a direct SOC measurement. The objective
of this study is to investigate, through numerical simulations, the bias of SOC estimates by
three different implementations of the EC tracer method. Hypothetic EC, OC, and $(OC/EC)_{pri}$
datasets with known break-down of POC and SOC values are numerically synthesized, then
SOC is estimated and compared with the "true" SOC as defined by the synthetic datasets. As
such, bias of SOC estimates using the various implementations of the EC tracer method can
be quantified.

**2 Evaluation of the Minimum R Squared Method**

**2.1 Data generation**

We first examine ambient OC and EC for the purpose of identifying distribution features that
can serve as the reference basis for parameterizing the numerical experiments. The one-year
hourly EC and OC measurement data from three sites in the PRD (one suburban site in
Guangzhou, a general urban site and a roadside site in Hong Kong, with more than 7000 data
at each site), are plotted in Figure S1 in the supplemental information (SI) document for the





whole year datasets and Figures S2-S4 for the seasonal subsets using the Nansha site as the

example. A brief account of the field ECOC analyzers and their field operation is provided in

the SI document. A detailed description of the measurement results and data interpretation for

the sites will be given in a separate paper. The distributions of measured OC, EC and OC/EC

are fitted by both normal and log-normal distribution curves and then examined by the

Kolmogorov–Smirnov (K-S) test. The K-S statistic, D, indicates that log-normal fits all three

distributions better than the normal distribution. Therefore, log-normal distributions are

adopted to define the OC, EC and OC/EC distributions during data generation in our

numerical experiments. Statistics of these ambient OC and EC, along with a few other

measurements reported in the literature, are summarized in Table 1 and are considered as the

reference for data generation to better represent the real situation.

The probability density function (PDF) for the log-normal distribution of variable x is:

$$f(x; \mu, \sigma) = \frac{1}{x\sigma\sqrt{2\pi}} \times e^{-\frac{(\ln(x)-\mu)^2}{2\sigma^2}} \qquad (3)$$

The two parameters, μ and σ, of the log-normal PDF are related to the average and standard

deviation of x through the following equations:

$$\mu = \ln(avg) - 0.5 \times \ln(1 + \frac{std^2}{avg^2}) \qquad (4)$$

$$\sigma = \sqrt{\ln(1 + \frac{std^2}{avg^2})} \qquad (5)$$

First, realistic average and standard deviation values of EC, (OC/EC)$_{pri}$, and OC (e.g. Figures

S1 – S5) are adopted to calculate μ and σ. Then pseudo random number generator use μ and σ

to synthesize EC and OC data sets.

The Mersenne twister (MT) (Matsumoto and Nishimura, 1998), a pseudorandom number

generator, is used in data generation. MT is provided as a function in Igor Pro. The system

clock is utilized as the initial condition for generation of pseudorandom numbers. The data

generated by MT has a very long period of $2^{19937} - 1$, permitting large data size and ensuring

that pseudorandom numbers are statistically independent between each data generation. The

latter feature ensures the independent relationship between EC and non-combustion related

SOC data. MT also allows assigning a log-normal distribution during pseudorandom number

generation to constrain the data. In a previous study, Chu (2005) used a variant of sine

functions to simulate POC and EC, which limited the data size to 120, and the frequency

distributions of POC and EC exhibited multiple peaks, a characteristic that is not realistic for


ambient measurements. The key information utilized in the EC tracer method is the correlation between EC and POC as well as the irrelevance between EC and SOC. The time series information is not needed in EC tracer method, making pseudorandom number generator a good fit for the evaluation purpose.

The procedure of data generation for the single emission source scenario is illustrated in Figure 2 and implemented by scripts written in Igor Pro. EC is first generated with the following parameters specified: sample size ($n$), average and relative standard deviation (RSD%) of the whole data set (see SI). The EC dataset statistically follows a log-normal distribution, while the sequence of each data point is randomly assigned. POC is then calculated by multiplying EC by $(OC/EC)_{pri}$ (Eq. 1). For simplicity, $(OC/EC)_{pri}$ is set to be a single value, while an analysis incorporating randomly generated log-normally distributed $(OC/EC)_{pri}$ values can be found in the SI material. SOC data is independently generated in a similar way to that for EC. The sum of POC and SOC then yields the synthesized OC. OC and EC data generated in this way are used to calculate SOC by different implementations of the EC tracer method. The bias of SOC estimation can then be evaluated by comparing the calculated SOC with the 'true' SOC values. Data generation for the scenarios with two primary emission sources is similar to the single source scenario and the steps are illustrated in Figure S6.

## 2.2 Scenario Study

Three scenarios are considered. Scenario 1 (S1) considers one single primary emission source. Scenario 2 (S2) considers two correlated primary emission sources, i.e., two sets of EC, POC, and each source has a single but different $(OC/EC)_{pri}$ value. An example of S2 is combined vehicular emissions from diesel-fuel and gasoline-fuel vehicles. These two sources of vehicular emissions have different $(OC/EC)_{pri}$, but often share a similar temporal variation pattern, making them well correlated. Scenario 3 (S3) considers two independent primary emission sources and simulates an ambient environment influenced by two independent primary emission sources, e.g. local vehicular emissions (lower $(OC/EC)_{pri}$) and regional biomass burning (higher $(OC/EC)_{pri}$).

In the following numerical experiments, three $(OC/EC)_{pri}$ estimation methods are examined and compared, including MRS, $OC/EC_{10\%}$ and $OC/EC_{min}$. As a single point, $OC/EC_{min}$, in ambient samples may be subjected to large random uncertainties, thus data with the lowest 1% OC/EC are adopted instead to derive the $OC/EC_{min}$.





### 2.2.1 Single primary source scenario

Both $OC/EC_{10\%}$ and $OC/EC_{min}$ methods rely on a subset of ambient OC and EC data to approximate $(OC/EC)_{pri}$. Figure 3 provides a conceptual illustration of the relationships
between $(OC/EC)_{pri}$ and the ambient OC/EC data, both are described to exhibit a log-normal distribution. As primary emissions move away from sources and aging processes start in the
atmosphere, SOC is added to the particle OC fraction, elevating OC/EC above $(OC/EC)_{pri}$. This in effect broadens the OC/EC distribution curve and shifts the distribution to the right
along the OC/EC axis, and the degree of broadening and shift depends on degree of aging process. The conventional EC tracer method using $OC/EC_{10\%}$ and $OC/EC_{min}$ assumes that the
left tail of ambient OC/EC distribution is very close to $(OC/EC)_{pri}$. This assumption, however, is fortuitous, rather than the norm. Two parameters, the distance between the means of the
$(OC/EC)_{pri}$ and ambient OC/EC distributions and the relative breadth of the two distributions, largely determines the closeness of the approximation of $OC/EC_{10\%}$ and $OC/EC_{min}$ to
$(OC/EC)_{pri}$. The distance between the two distributions depends on the fraction of SOC in OC (i.e., $f_{SOC}$), while the width of the ambient OC/EC distribution is closely associated with RSD
of SOC ($RSD_{SOC}$) and the width of the $(OC/EC)_{pri}$ distribution is reflected in $RSD_{POC}$ and $RSD_{EC}$. As shown in Figure 3a, only an appropriate combination of distance of the two
distribution means and variances could lead to a close approximation of the $(OC/EC)_{pri}$ by $OC/EC_{10\%}$ or $OC/EC_{min}$ (i.e., the left tail of OC/EC distribution). If the ambient aerosol has a
significant $f_{SOC}$ shifting the ambient OC/EC distribution such that its left tail is beyond $(OC/EC)_{pri}$ (Figure3b), then the left tail would overestimate $(OC/EC)_{pri}$. Underestimation of
$(OC/EC)_{pri}$ could also happen in theory as shown in Figure 3c if the ambient minimum OC/EC (left tail) is less than the mean of the $(OC/EC)_{pri}$ distribution (i.e., under conditions of
very small $f_{SOC}$).

   The above analysis reveals $f_{SOC}$, $RSD_{SOC}$, $RSD_{POC}$, and $RSD_{EC}$ are key parameters in

influencing the accuracy of SOC estimation. As a result, they are chosen in the subsequent sensitivity tests in probing the SOC estimate bias under conditions of different carbonaceous
aerosol compositions.

   SOC estimation bias in S1 as a function of $RSD_{SOC}$ and $RSD_{EC}$ are shown in Figures 4a and

4b. The SOC estimate by MRS is not affected by the magnitude of $RSD_{EC}$ and $RSD_{SOC}$, and is in excellent agreement with the true values (Figure 4). In comparison, SOC by $OC/EC_{10\%}$
and $OC/EC_{min}$ is consistently biased lower and the degree of negative bias becomes larger with decreasing $RSD_{SOC}$ or $RSD_{EC}$. The $OC/EC_{10\%}$ method always produces larger negative





bias than the $OC/EC_{min}$ method. At $RSD_{SOC}$ and $RSD_{EC}$ at 50%, SOC estimate has a -14%

bias by $(OC/EC)_{min}$ and a -45% bias by $OC/EC_{10\%}$. These results confirm the hypothesis

illustrated in the conceptual diagram (Figure 3) that the validity of using the left tail of

OC/EC distribution depends on the distance of its distribution mean from $(OC/EC)_{pri}$ and the

distribution breadth. Both $OC/EC_{10\%}$ and the $OC/EC_{min}$ methods underestimate SOC and the

degree of underestimation by the $OC/EC_{10\%}$ method is worse.

**2.2.2 Scenarios assuming two primary sources**

In the real atmosphere, multiple combustion sources impacting a site is normal. We next

evaluate the performance of the MRS method in scenarios of two primary sources and

arbitrarily dictate that the $(OC/EC)_{pri}$ of source 1 is lower than source 2. Common

configurations in S2 and S3 include: $EC_{total}=2\pm0.4$ $\mu gm^{-3}$; proportion of source 1 EC to total

EC ($f_{EC1}$) varies from 0 to 100%; ratio of the two $OC/EC_{pri}$ values ($\gamma_{\_pri}$) vary in the range of

2~8.

In Scenario 2 (i.e., two correlated primary sources), three factors are examined, including $f_{EC1}$,

$\gamma_{\_pri}$ and $f_{SOC}$, to probe their effects on SOC estimation. By varying $f_{EC1}$, the effect of

different mixing ratios of two sources can be examined, as $f_{EC1}$ is expected to vary within the

same ambient dataset as a result of spatiotemporal dynamics of air masses. MRS reports

unbiased SOC, irrespective of different $f_{EC1}$ and $f_{SOC}$ or $\gamma_{\_pri}$ (Figure 5). In comparison, SOC

by $OC/EC_{10\%}$ and $OC/EC_{min}$ are underestimated. The degree of underestimation depends on

$f_{SOC}$, e.g., -12% at $f_{SOC} = 25\%$ versus -20% at $f_{SOC} =40\%$ in the $OC/EC_{min}$ method while the

magnitude of underestimation has a very weak dependence on $f_{SOC}$ in the $OC/EC_{10\%}$ method,

staying around -40 % as $f_{SOC}$ is doubled from 20% to 40%. The degree of SOC bias by

$OC/EC_{10\%}$ and $OC/EC_{min}$ are independent of $f_{EC1}$ and $\gamma_{\_pri}$, as SOC bias is associated with

$RSD_{EC}$, $RSD_{SOC}$ and $f_{SOC}$. Since two primary sources are well correlated, $RSD_{EC}$ is equivalent

between the two sources. As a result, the overall $RSD_{EC}$ is constant when $f_{EC1}$ and $\gamma_{\_pri}$ vary,

and the SOC bias is independent of $f_{EC1}$ and $\gamma_{\_pri}$

In summary, in scenarios of two well-correlated primary combustion sources, MRS always

produces unbiased SOC estimates while $OC/EC_{min}$ and $OC/EC_{10\%}$ consistently underestimate

SOC, with $OC/EC_{10\%}$ producing larger negative bias.

As for Scenario 3 in which two independent primary sources co-exist, SOC estimates by

MRS could be biased and the degree and direction of bias depends on $f_{EC1}$. Figure 6a shows

the variation of SOC bias with $f_{EC1}$ when $f_{SOC}$ is fixed at 40%. The variation of SOC bias by




MRS with $f_{EC1}$ follows a pseudo-sine curve, exhibiting negative bias when $f_{EC1} < 50\%$ (i.e.,
EC is dominated by source 2, the higher $(OC/EC)_{pri}$ source) and positive bias when $f_{EC1} > 50\%$

and the range of bias are confined to -20% to -40% under the condition of $f_{SOC}=40\%$. In
comparison, the $OC/EC_{min}$ and $OC/EC_{10\%}$ methods again consistently underestimate SOC by

more than -50%, with the bias worsened in the $OC/EC_{10\%}$ method.

The bias variation range becomes narrower with increasing $f_{SOC}$ in the MRS method, as

shown by the boxplots for four $f_{SOC}$ conditions (20%, 40%, 60%, and 80%) in Figure 6b. The
MRS-derived SOC bias range is reduced from -20–+40% at $f_{SOC} = 40\%$ to -10–+20% at $f_{SOC}$

= 60%, further to -6–+10% at $f_{SOC} = 80\%$. In the other two methods, the SOC bias does not
improve with increasing $f_{SOC}$. Dependence of the SOC estimation bias on $\gamma_{\_pri}$ is examined in

Figure 6c showing the higher $\gamma_{\_pri}$ induces a higher amplitude of the SOC bias. If OC is
dominated by SOC (e.g., $f_{SOC} =80\%$), SOC bias by MRS is within 10%.

A variant of MRS implementation (denoted as MRS') is examined, with the important
difference that $EC_1$ and $EC_2$, attributed to source 1 and source 2, respectively, are used as

inputs instead of total EC. With the knowledge of EC breakdown between the two primary
sources, $(OC/EC)_{pri1}$ can be determined by MRS from $EC_1$ and $OC_{total}$. Similarly $(OC/EC)_{pri2}$

can be calculated by MRS from $EC_2$ and $OC_{total}$. SOC is then calculated with the following
equation:

$$SOC = OC_{total} - (OC/EC)_{pri1} \times EC_1 - (OC/EC)_{pri2} \times EC_2 \qquad (6)$$

MRS' produces unbiased SOC, irrespective of the different carbonaceous compositions

(Figure 6). However, we note that there is a great challenge in meeting the data needs of
MRS' as $EC_1$ and $EC_2$ are not available.

In scenario 3, the simulation results imply that three factors are associated with the SOC bias
by MRS, including: $f_{EC1}$, $\gamma_{\_pri}$ and $f_{SOC}$. The first factor controls whether SOC bias by MRS is

positive or negative. The latter two affect the degree of SOC bias. For high $f_{SOC}$ conditions,
the bias could be acceptable. If $EC_1$ and $EC_2$ can be differentiated for calculating individual

$(OC/EC)_{pri}$ of each source, unbiased SOC estimation is achievable regardless of what values
$f_{EC1}$, $\gamma_{\_pri}$ and $f_{SOC}$ take.

**2.3 Impact of measurement uncertainty**

In the preceding numerical analysis, the simulated EC and OC are not assigned any

measurement uncertainty; however, in reality, every EC and OC measurement is associated





with a certain degree of measurement uncertainty. We next examine the influence of OC and

EC measurement uncertainty on SOC estimation accuracy by different EC tracer methods. The uncertainties are assumed to follow a uniform distribution and generated separately by
MT. It is also assumed that the uncertainty ( $\varepsilon_{EC}$ or $\varepsilon_{OC}$ ) is proportional to the concentration of EC and OC through the multiplier $\gamma_{unc}$ (i.e., relative measurement uncertainty).
$$-\gamma_{unc}EC \leq \varepsilon_{EC} \leq \gamma_{unc}EC \qquad (7)$$

$$-\gamma_{unc}OC \leq \varepsilon_{OC} \leq \gamma_{unc}OC \qquad (8)$$

The measurement uncertainties of POC and SOC are then back-calculated following the uncertainty propagation formula and assuming the same relative measurement uncertainty for
POC and SOC (Harris, 2010)

$$\gamma'_{unc} = \gamma_{unc}\sqrt{\frac{OC^2}{POC^2+SOC^2}} \qquad (9)$$

$$-\gamma'_{unc}POC \leq \varepsilon_{POC} \leq \gamma'_{unc}POC \qquad (10)$$

$$-\gamma'_{unc}SOC \leq \varepsilon_{SOC} \leq \gamma'_{unc}SOC \qquad (11)$$

The simulated EC, POC and SOC with measurement uncertainties (abbreviated as EC$_{simulated}$, POC$_{simulated}$ and SOC$_{simulated}$ respectively) are determined as:
$$EC_{simulated} = EC_{true} + \varepsilon_{EC} \qquad (12)$$

$$POC_{simulated} = POC_{true} + \varepsilon_{POC} \qquad (13)$$

$$SOC_{simulated} = SOC_{true} + \varepsilon_{SOC} \qquad (14)$$

Sensitivity tests of SOC estimation as a function of relative measurement uncertainty ($\gamma_{unc}$)

and f$_{SOC}$ is performed as shown in Figure 7. Fixed input parameters include: N=8000; EC = 2±1 $\mu$gm$^{-3}$; (OC/EC)$_{pri}$ = 0.5. Studies by Chu (2005) and Saylor et al. (2006) both suggest
ratio of average POC to average EC (ROA, see SI for details) is the best estimator of the expected primary OC/EC ratio because it is mathematically equivalent to the true regression
slope when the data contains no intercept. ROA is confirmed as the best representation of (OC/EC)$_{pri}$ for SOC estimation, which shows no bias towards $\gamma_{unc}$ or f$_{SOC}$ change. MRS
overestimates SOC and the positive bias increases with $\gamma_{unc}$ while decreases with f$_{SOC}$ (Figure 7). The SOC estimates by OC/EC$_{min}$ and OC/EC$_{10\%}$ exhibit larger bias than those by
MRS. For example, as shown in Figure 7a, when f$_{SOC}$ =20% and $\gamma_{unc}$ = 10%, the bias of SOC by MRS, OC/EC$_{10\%}$ and OC/EC$_{min}$ is 8%, -28% and 36%, respectively. With increasing f$_{SOC}$,





the bias of SOC by OC/EC$_{min}$ decreases while the bias of SOC by OC/EC$_{10\%}$ increases when

$\gamma_{unc}$ = 10-20%. MRS always demonstrates the best performance in SOC determination

amongst the three (OC/EC)$_{pri}$ estimation methods. When $\gamma_{unc}$ could be controlled within 20%,

the SOC bias by MRS does not exceed 23% when f$_{SOC}$=20%. If the f$_{SOC}$ ratio falls in the

range of 60-80% and $\gamma_{unc}$ is <20%, the OC/EC$_{min}$ has a similar performance as MRS, but

SOC by OC/EC$_{10\%}$ still shows a large bias (~41%) (Figures 7c and 7d).

Sensitivity studies of SOC estimation as a function of $\gamma_{unc}$ and (OC/EC)$_{pri}$ are performed and

the results are shown in Figure S7. In all the three (OC/EC)$_{pri}$ representations, SOC estimates

are sensitive to $\gamma_{unc}$ but insensitive to the magnitude of (OC/EC)$_{pri}$. In the single primary

source scenario (S1), it is proved that the performance of MRS regarding SOC estimation is

mainly affected by $\gamma_{unc}$ and to a less degree by f$_{SOC}$. Other variables such as (OC/EC)$_{pri}$ and

EC concentration do not affect the accuracy of SOC estimation.

**3 Caveats of the MRS method in its applications to ambient data**

Table 2 summarizes the performance in terms of SOC estimation bias by the different

implementations of the EC tracer method, assuming typical variation characteristics for

ambient ECOC data. When employing the EC tracer method on ambient samples, it is clear

that MRS is preferred since it can provide more accurate SOC estimation.

If the sampling site is dominated by a single primary source (similar to Scenario 1), MRS can

perform much better than the traditional OC/EC percentile and minimum approaches. Two

issues should be paid attention to when applying MRS: (1) MRS relies on the irrelevance of

EC and SOC. This assumption could be invalid if a fraction of SOC is formed from semi-

volatile POC (here referred as SOC$_{svP}$) (Robinson et al., 2007). Since POC is well correlated

with EC, this SOC$_{svP}$ would be attributed to POC by MRS, causing SOC underestimation.

The interference of SOC$_{svP}$ will be discussed in a separate paper. (2) OC$_{non-comb}$ will be

attributed to SOC if only EC is used as a tracer. If OC$_{non-comb}$ is small compared to SOC, such

approximation is acceptable. Otherwise quantification of its contribution is needed. If a stable

tracer for OC$_{non-comb}$ is available, determination of OC$_{non-comb}$ contribution by MRS is possible,

since this scenario is mathematically equivalent to S3 (e.g., relabel EC2 to tracer of OC$_{non-}$

$_{comb}$ and POC to OC$_{non-comb}$).

If the sampling site is influenced by two correlated primary sources with distinct (OC/EC)$_{pri}$

(Scenario 2, e.g. urban areas that have vehicular emission from both gasoline and diesel),

MRS is still much more reliable than the traditional OC/EC percentile and minimum





approaches. If the sampling site is influenced by two independent primary sources with distinct $(OC/EC)_{pri}$ (Scenario 3, e.g. vehicular emission and biomass burning), SOC
estimation by MRS is better than the other two conventional methods. But it should be noted that possible bias may exist and the magnitude of bias depends on the relative abundance
between the two sources. If tracers are available to demarcate the EC contributions by the different primary sources, unbiased SOC estimation is possible by employing these tracers in
MRS.

## 4 Application of the MRS method to low time-resolution ECOC data

Besides hourly measurements of EC and EC by online aerosol carbon analyzers, the MRS method could also be applied to offline measurements of OC and EC based on filters
collected over longer durations (i.e., 24 h), which are more readily available around the world. To explore the impact of sampling duration (e.g., hourly vs. daily), we here use one-year
hourly data at the suburban site of Guangzhou to average them into longer intervals of 2-24 h. The 24 h-averaged samples yield a $(OC/EC)_{pri}$ of 2.53, 12% higher than the $(OC/EC)_{pri}$
derived from hourly data (2.26). The is the result of that OC/EC distributions are narrowed when the averaging interval lengthens (Figure 8), leading to elevation of the MRS-derived
$(OC/EC)_{pri}$. As many $PM_{2.5}$ speciation networks adopt a sampling schedule of one 24-h sample every six days, we further extract the every-six-day samples to do the MRS
calculation. The one-year data yields six subsets of every-six-day samples, and the MRS calculation produces the $OC/EC_{pri}$ in the range of 2.37 – 2.75 (5-22% higher than the
$OC/EC_{pri}$ from the hourly data). This example illustrates that if 24-h sample ECOC data are used, SOC would be biased slightly lower in comparison with those derived from the hourly
data.

## 5 Conclusions

In this study, the accuracy of SOC estimation by EC tracer method is evaluated by comparing three $(OC/EC)_{pri}$ determination approaches using numerically simulated data. The MRS
method has a clear quantitative criterion for the $(OC/EC)_{pri}$ calculation, while the other two commonly used methods, namely minimum OC/EC $(OC/EC_{min})$ and OC/EC percentile (e.g.
$OC/EC_{10\%})$, are empirical in nature. Three scenarios are considered in the numerical simulations to evaluate the SOC estimation bias by the different EC tracer methods assuming
typical variation characteristics for ambient ECOC data. In the scenarios of a single primary source and two well-correlated primary combustion sources, MRS always produces unbiased



SOC estimates while $OC/EC_{min}$ and $OC/EC_{10\%}$ consistently underestimate SOC. In the scenario of two independent primary sources, SOC by MRS exhibit bias but still perform
better than $OC/EC_{min}$ and $OC/EC_{10\%}$. If EC from each independent source can be differentiated to allow calculation of individual $(OC/EC)_{pri}$ for each source, unbiased SOC
estimation is achievable. Sensitivity tests of OC and EC measurement uncertainty on SOC estimation demonstrate the superior accuracy of MRS over the other two approaches. It is
clear that when employing the EC tracer method to estimate SOC, MRS is preferred over the two conventional methods ($OC/EC_{10\%}$ and $OC/EC_{min}$) since it can provide more accurate
SOC estimation. We also evaluated the impact of longer sampling duration on derived $(OC/EC)_{pri}$ and found that if 24-h sample ECOC data are used, SOC would be biased slightly
lower in comparison with those derived from the hourly data.
**Supporting Information**

    Experimental methods and additional figures. This material is available free of charge via the
Internet.
**Acknowledgements**

    This work is supported by the National Science Foundation of China (21177031), and the
Fok Ying Tung Graduate School (NRC06/07.SC01). The authors thank Hong Kong Environmental Protection Department for making available the ECOC data at Tsuen Wan and
Prof. Dui Wu of Institute of Tropical and Marine Meteorology, China Meteorological Administration for providing logistic support of OC EC measurements in Nancun. The
authors are also grateful to Dr. Stephen M Griffith for the helpful comments.





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





**Table 1.** Summary of statistics of OC and EC in ambient samples

| Location | Site Type | Sampling Period | Time resolution | $RSD_{EC}$ (%) | $RSD_{SOC}$ (%) | SOC estimation method | $f_{SOC}$ mass fraction (%) | | | Ref |
|---|---|---|---|---|---|---|---|---|---|---|
| | | | | | | | Avg | Min | Max | |
| Hong Kong, PRD | Suburban | July 2006, local days July 2006, regional days | 24 hr | | | PMF | 25% 65% | 6% 46% | 79% 89% | (Hu et al., 2010) |
| Hong Kong, PRD | Urban | May 2011 – Apr. 2012 | 1 hr | 51% | | EC tracer PMF | | | | (Huang et al., 2014) |
| Guangzhou, PRD | Rural | July 2006 | 1 hr | 154% | 115% | EC tracer | 47% | | 80% | (Hu et al., 2012) |
| Guangzhou, PRD | Suburban | Feb 2012 – Jan 2013 | 1 hr | 86% | 84% | EC tracer | 41% | 0% | 86% | This study |
| Beijing | Urban | Winter 2005 Spring 2006 Summer 2006 Fall 2006 | 1 hr | | | EC tracer | 19% 27% 45% 23% | | | (Lin et al., 2009) |
| Pittsburgh | Suburban | Jul. 2001 – Aug. 2002 | 2-4 hr | | | EC tracer | 38% | | | (Polidori et al., 2006) |
| Mt. Tai, China | Rural | Mar. – Apr. 2007 Jun. – Jul. 2007 | 1 hr | 89% 69% | | EC tracer | 60% 73% | | | (Wang et al., 2012) |
| Jeju Island, Korea | Rural | May – Jun. 2009 Aug – Sep 2009 | 1 hr | 53% 57% | 117% 102% | EC tracer | 31% 18% | | | (Batmunkh et al., 2011) |



**Table 2.** Summary of numerical study results under different scenarios [a].

| | Tested parameter | SOC bias | | | |
| --- | --- | --- | --- | --- | --- |
| | | MRS[b] | MRS'[c] | $OC/EC_{min}$ | $OC/EC_{10\%}$ |
| Scenario 1 Single source | $RSD_{EC}$ | ±4% | | -13% ~ -7% | -43% ~ -36% |
| | $RSD_{SOC}$ | ±4% | | -11% ~ -4% | -42% ~ -22% |
| | $\gamma_{unc}$ | +10% | | -12% ~ 20% | -43% ~ -32% |
| Scenario 2 Two correlated sources | $f_{EC1}$ | ±4% | | -20% | -40% |
| | $\gamma\_pri$ | ±4% | | -20% | -40% |
| | $f_{SOC}$ | ±4% | | -20% | -40% |
| Scenario 3 Two independent sources | $f_{EC1}$ | -20%~40% | ±10% | -50% | -60% |
| | $\gamma\_pri$ | -20%~40% | ±10% | -50% | -60% |
| | $f_{SOC}$ | -20%~40% | ±10% | -50% | -60% |

[a] Results shown here are obtained assuming the following ambient conditions: $RSD_{EC}$ 50-100%; $f_{SOC}$ 40-60%; $\gamma_{unc}$ 20%;
[b] "+" represents SOC overestimation and "-" represents underestimation;
[c] MRS': In S3, EC1 and EC2 are used for SOC calculation.



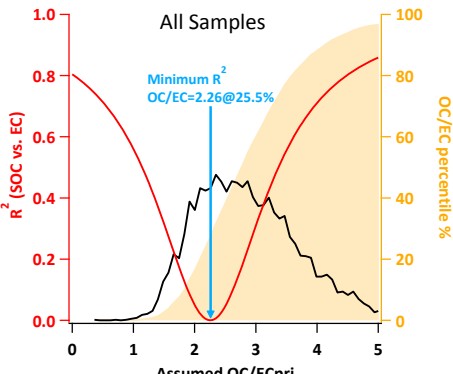

**Figure 1**. Illustration of the minimum R square method (MRS) to determine $OC/EC_{pri}$ using one year of hourly OC and EC measurements at a suburban site in the Pearl River Delta, China. The red curve shows the correlation coefficient ($R^2$) between SOC and EC as a function of assumed $OC/EC_{pri}$. The black curve is the frequency distribution of the OC/EC ratio for the entire OC and EC data set. The shaded area in tan represents the cumulative frequency curve of OC/EC ratio.





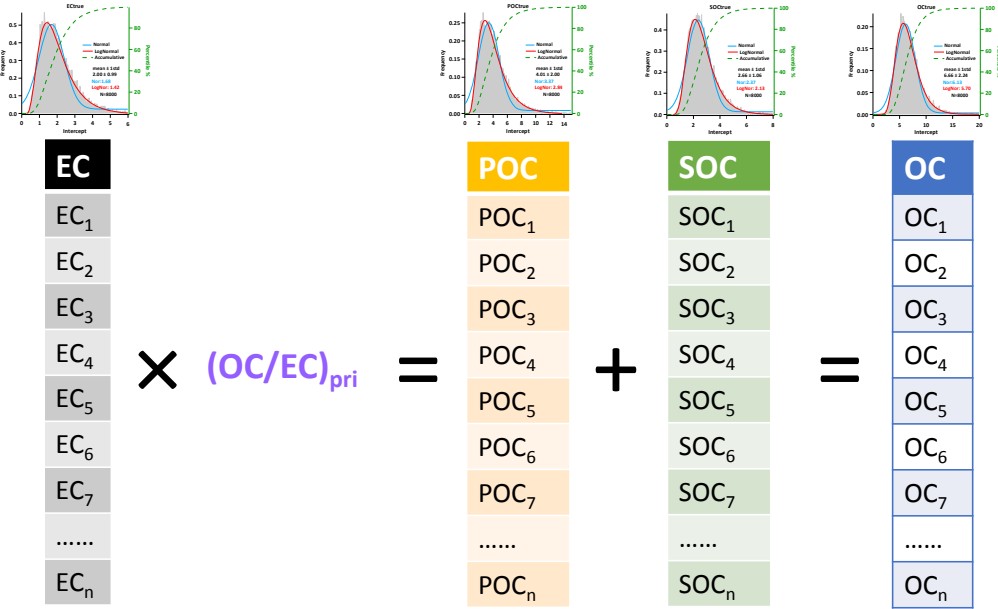

**Figure 2.** Schematic diagram of pseudorandom number generation for the single emission source scenario that assumes (OC/EC)$_{pri}$ is a single value. The data series (EC and SOC), generated by Mersenne twister (MT) pseudorandom number generator, statistically follow a log-normal distribution, but the sequence of each data point is randomly assigned.





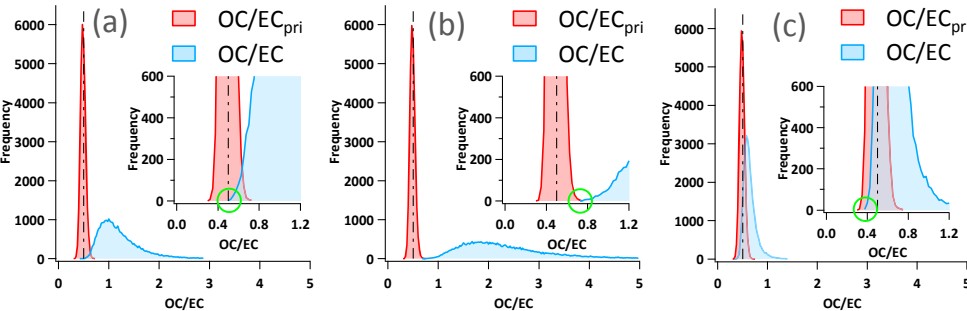

**Figure 3.** Conceptual diagram illustrating three scenarios of the relationship between $(OC/EC)_{pri}$ and ambient OC/EC measurements. Both are assumed to be log-normally distributed. (a) Ambient minimum (left tail) is equal to the peak of $(OC/EC)_{pri}$. (b) Ambient minimum OC/EC (left tail) is larger than the mean of $(OC/EC)_{pri}$. (c) Ambient minimum OC/EC (left tail) is less than the peak of $(OC/EC)_{pri}$.





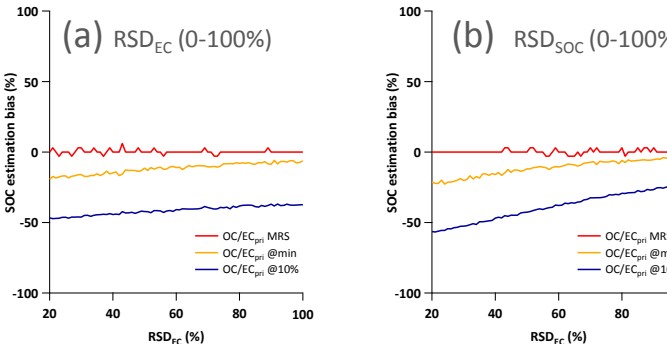

**Figure 4.** Bias of SOC determination as a function of: (a) $RSD_{EC}$; (b) $RSD_{SOC}$. Different representation of $(OC/EC)_{pri}$ include: ratio of averages (ROA), MRS, $OC/EC_{min}$ and $OC/EC_{10\%}$.. Fixed input parameters: N = 8000; EC = 2±1 µgC m$^{-3}$; $(OC/EC)_{pri}$ = 0.5; POC = 1 ±0.5 µgC m$^{-3}$, $f_{SOC}$ =40%, and SOC = 0.67±0.34 µgC m$^{-3}$.





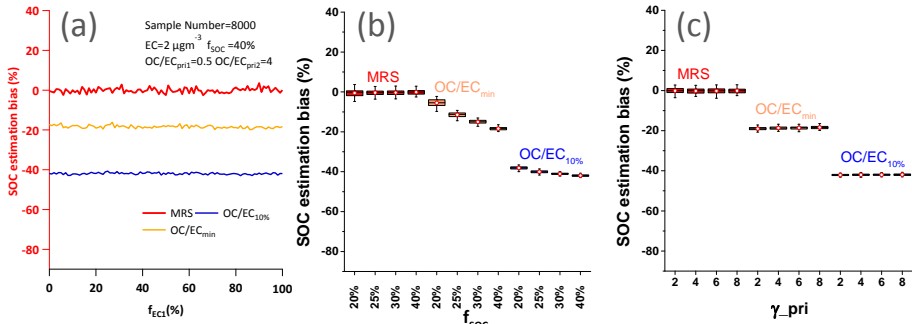

**Figure 5.** SOC bias in Scenario 2 (two correlated primary emission sources of different $(OC/EC)_{pri}$) as estimated by four different EC tracer methods denoted in red, blue and yellow. (a) SOC bias as a function of $f_{EC1}$. Results shown here are calculated using $f_{SOC} = 40\%$ as an example. (b) Range of SOC bias shown in boxplots for four $f_{SOC}$ conditions (20%, 25%, 30% and 40%). (c) Range of SOC bias shown in boxplots for four $\gamma\_pri$ conditions (2, 4, 6 and 8). The symbols in the boxplots are white circles for average, the line inside the box for median, the box boundaries representing the 75th and the 25th percentile, and the whiskers representing the 95th and 5th percentile.





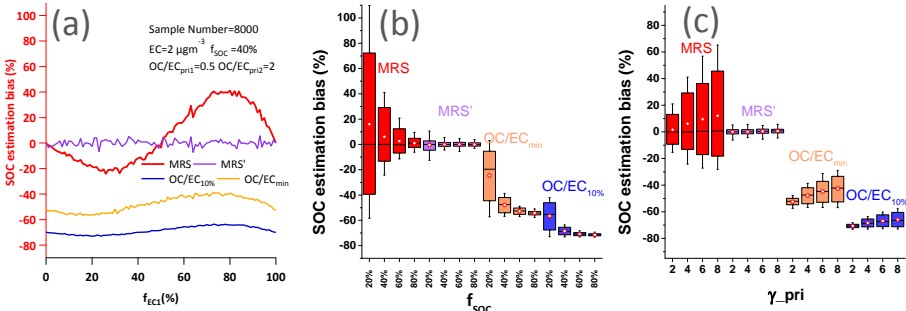

**Figure 6.** SOC bias in Scenario 3 (two independent primary emission sources of different $(OC/EC)_{pri}$) as estimated by four different EC tracer methods denoted in red, purple, blue and yellow. MRS' differs from MRS in that $EC_1$ and $EC_2$ instead of total EC are used as inputs. (a) SOC bias as a function of $f_{EC1}$. Results shown here are calculated using $f_{SOC} = 40\%$ as an example. (b) Range of SOC bias shown in boxplots for four $f_{SOC}$ conditions (20%, 40%, 60% and 80%). (c) Range of SOC bias shown in boxplots for four $\gamma\_pri$ conditions (2, 4, 6 and 8). The symbols in the boxplots are white circles as average, the line inside the box as median, upper and lower boundaries of the box representing the 75th and the 25th percentile, and the whiskers above and below each box representing the 95th and 5th percentile.





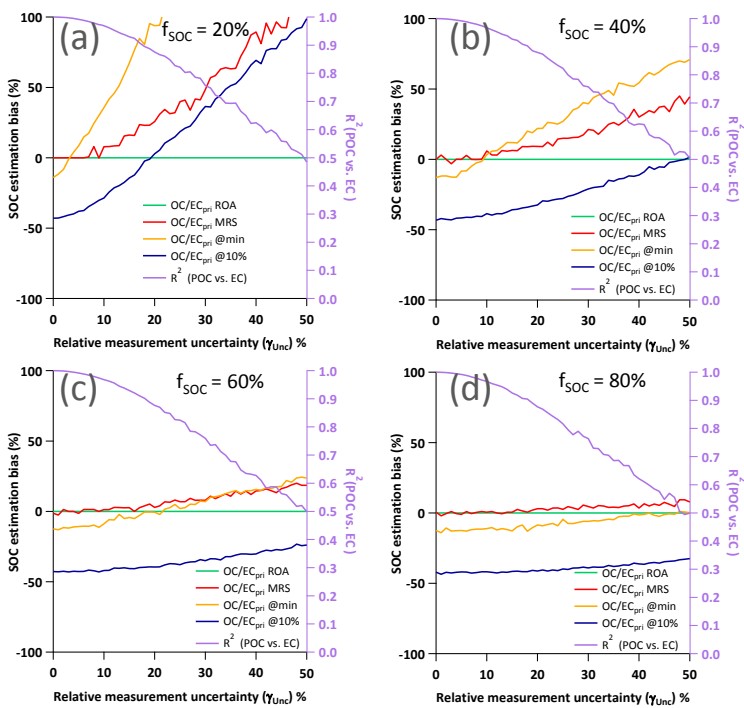

**Figure 7.** Bias of SOC determination as a function of relative measurement uncertainty ($\gamma_{unc}$) and SOC/OC ratio ($f_{SOC}$) by different approaches of estimating $(OC/EC)_{pri}$, including ratio of averages (ROA), minimum R square (MRS), $OC/EC_{10\%}$, and $OC/EC_{min}$. Fixed input parameters: N=8000; EC = 2±1 μgm$^{-3}$; $(OC/EC)_{pri}$ = 0.5. Variable input parameters: (a) $f_{SOC}$ =20%, SOC = 0.25±0.13 μgC m$^{-3}$, (b) $f_{SOC}$ =40%, SOC = 0.67±0.33 μgC m$^{-3}$, (c) $f_{SOC}$ =60%, SOC = 1.5±0. 75 μgC m$^{-3}$, and (d) $f_{SOC}$ =80%, SOC = 4±2 μgC m$^{-3}$





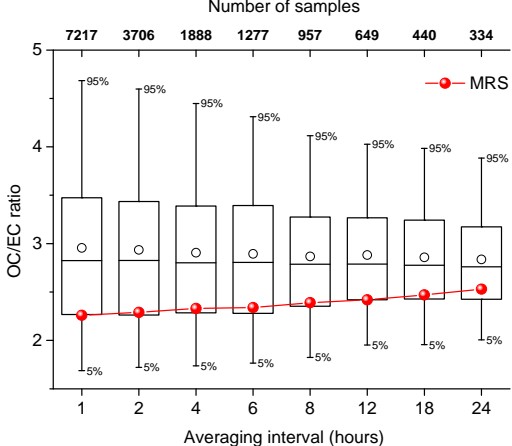

**Figure 8.** OC/EC distributions assuming different average intervals from 2 to 24 h and the corresponding MRS-derived OC/EC$_{pri}$. The bottom x-axis represents averaging interval (e.g. 1 h is the original data time resolution, 2 h referring average 1-h data into 2-h interval data, etc). The top x-axis represents the number of data point corresponding to the respective data averaging interval. Distributions of OC/EC ratio at various averaging intervals are shown as box plots (Empty circles: average, the line inside the box: median, the box boundaries: 75$^{th}$ and the 25$^{th}$ percentile, and the whiskers: 95$^{th}$ and 5$^{th}$ percentile). The red dots represent calculated (OC/EC)$_{pri}$ by MRS.