# Peer review of "Determination of Primary combustion source organic carbon-to-elemental carbon (OC/EC) ratio using ambient OC and EC measurements: Secondary OC-EC correlation minimization method"

_Atmospheric Chemistry and Physics, 2015_

## Referee Comment (RC1) · Anonymous Referee #1 · 9 Feb 2016

Dear Editor, this MS presents a statistical assessment of an alternative method to quantify secondary organic carbon (SOC) in ambient air samples. This method is an alternative to the classic EC tracer method. It is a useful assessment of an alternative method which seems to perform rather well, and therefore merits publication. Reading is somewhat complicated due to the frequent use of abbreviations (eg, fSOC), though. A more fluent writing style would help the reader.

Some specific comments:
[Figure]

- line 75: I believe Pio et al propose yet another method, using a subset of samples with 5% lowest ratios and discarding the 3 lowest... I don't have the exact reference right now, but please add. - line 90: any reason why the Millet method was overlooked? - line 211: please elaborate on why the OCEC10% method provides worse results - line 226: I don't understand the different behavior of the OCEc10% amend the OCeCmin methods, given that they are both subsets of the total dataset with specific characteristics of representing 1% and 10%. Why is their behavior different? - section uncertainty: with some analytical methods (e.g., TOT) the uncertainty is mostly constant (0,1-0,2 micrograms/cm2), please discuss how this would affect the results in this section. - line 317, please clarify what the authors mean by "the irrelevance of EC and SOC", it is unclear to me

---

## Referee Comment (RC2) · Anonymous Referee #2 · 17 Mar 2016

**Generally Comments**

Typically the EC tracer method, when used in estimating the secondary organic carbon (SOC), relies on three conditions– 1) the relatively constant  $(OC/EC)_{pri}$  over the period of study; 2) the random nature of SOC formation relative to EC; and 3) a subset of dataset without significant SOC contributions. The  $OC/EC_{10\%}$  or  $OC/EC_{min}$  essentially utilize the subset in Condition #3 to derive the  $(OC/EC)_{pri}$  if it does have an unique value. Any deviations from the conditions as well as measurement uncertainties will

lead to bias in determining  $(OC/EC)_{pri}$ . In some environments where SOC dominates, the third condition is generally impossible to be met. This study, through an extensive test, shows that the third condition is not necessary in calculating  $(OC/EC)_{pri}$ , if an algorithm, i.e., minimum R2 (MRS), is used looking for  $(OC/EC)_{pri}$  that yields SOC least correlated with EC. Without further examinations, the reviewer thinks that MRS is probably mathematically rigorous for any datasets satisfying the first two conditions and, additionally, with sufficient size and accuracy. It can perform better than  $OC/EC_{10\%}$  or  $OC/EC_{min}$  most of the time because Condition 3 is fortuitous, as described by the authors.

While the reviewer agrees that MRS should be used instead of  $OC/EC_{10\%}$  or  $OC/EC_{min}$  in calculating SOC, particularly for a large dataset which can support meaningful correlation analysis, MRS does not solve fundamental problems in the EC tracer method. The  $(OC/EC)_{pri}$  is by no means constant, as it varies with source contributions from day to day and season to season. SOC is likely correlated with EC because in urban areas many SOC precursors originate from the same combustion sources as EC. This paper demonstrates that when Conditions 1 and 2 are in doubt, MRS produces erroneous results. MRS results are also sensitive to measurement uncertainty that impacts the correlation coefficients. These limitations, however, are not emphasized adequately in the abstract, which sounds almost like MRS has tackled all these issues. These issues, still, can only be solved by using multivariate or chemical mass balance analysis with additional markers.

**Specific Comments**

Abstract: Please describe the assumptions of MRS, datasets that are suitable for MRS analysis, and potential errors while in the same time shortening the abstract. Just saying MRS is better than OC/EC10% or OC/ECmin is not meaningful because all the three could be very wrong in some cases.

Line 97-102: While using simulated data is insightful, it offers no proof. The authors
may explore if there is a true "proof" from mathematical or statistical derivations that MRS will yield true  $(OC/EC)_{pri}$  if SOC is indeed random and the dataset is big enough. This may also answer the question- how big is big? MRS does not seem suitable for a dataset with only dozens of points.

Line 116-118: How good are the K-S statistics? In other words, how well did the pseudorandom number generator reproduce the statistics in the original dataset?

Line 126: Eqs. (4)-(5) do not work for all datasets. They are probably asymptotes when datasets are large enough in size.

Line 136: Mention here that the case with combustion-related SOC is discussed later.

Line 151-152: The results of log-normally distributed  $(OE/CC)_{pri}$  should be summarized in the text if possible.

Line 220-222: It is not clear if  $f_{EC1}$  was varied from sample to sample in a single test or only varied from test to test. If the former, how could you make sure EC1 and EC2 are highly correlated?

Line 284-286: Since POC and SOC are not directly measured, what is the meaning to simulate their measurement uncertainty?

Line 384: How were the six subsets selected?

Line 360-362: Emphasize that this only happens when measurement uncertainties are small.

**ACPD**

---

## Author Response (AR1)

**Point-by-point response to review comments on manuscript acp-2015-997**
**"Determination of Primary combustion source organic carbon-to-elemental carbon (OC/EC) ratio using ambient OC and EC measurements: Secondary OC-EC correlation minimization method"**

**By Cheng Wu and Jian Zhen Yu**

We thank the two anonymous reviewers for their constructive comments. Our point-by-point responses to the review comments are listed below. Changes to the manuscript are marked in blue in the revised manuscript. The marked manuscript is submitted together with this response document.

**Anonymous Referee #1**

Dear Editor, this MS presents a statistical assessment of an alternative method to quantify secondary organic carbon (SOC) in ambient air samples. This method is an alternative to the classic EC tracer method. It is a useful assessment of an alternative method which seems to perform rather well, and therefore merits publication. Reading is somewhat complicated due to the frequent use of abbreviations (eg, fSOC), though. A more fluent writing style would help the reader.

**Author's Response:** We add a table (also shown below) in the revised main text to help readers to have a quick check of abbreviations used in the paper. We believe this would be more reader-friendly than looking for definitions that scattered in the main text. Please see below for point-by-point response to reviewers' comments.

**Table 1.** Acronyms and Abbreviations

| Abbreviation | Definition |
| --- | --- |
| EC | elemental carbon |
| $EC_1$, $EC_2$ | EC from source 1 and source 2 in the two-source scenario |
| $f_{EC1}$ | fraction of EC from source 1 to the total EC |
| $f_{SOC}$ | ratio of SOC to OC |
| MRS | minimum R squared method |
| MRS' | A variant of MRS that use EC from individual sources as input |
| MT | Mersenne twister pseudorandom number generator |
| $n$ | sample size in MT data generation |
| OC | organic carbon |
| OC/EC | OC to EC ratio |
| $(OC/EC)_{pri}$ | primary OC/EC |
| $OC/EC_{10\%}$ | OC/EC at 10% percentile |
| $OC/EC_{min}$ | minimum OC/EC |
| $OC_{non\text{-}comb}$ | OC from non-combustion sources |
| PDF | probability density function of a distribution |
| POC | primary organic carbon |
| ROA | ratio of averages |
| RSD | relative standard deviation |
| $RSD_{EC}$ | RSD of EC |
| $RSD_{POC}$ | RSD of POC |
| $RSD_{SOC}$ | RSD of SOC |
| SOC | secondary organic carbon |
| $SOC_{svP}$ | SOC formed from semi-volatile POC |
| $\Upsilon\_pri$ | ratio of the $(OC/EC)_{pri}$ of source 2 to source 1 |
| $\varepsilon_{EC}$, $\varepsilon_{OC}$ | measurement uncertainty of EC and OC |
| $\Upsilon_{unc}$ | relative measurement uncertainty |
| $\gamma\_{RSD}$ | the ratio between the RSD values of $(OC/EC)_{pri}$ and EC |

Some specific comments:

- line 75: I believe Pio et al propose yet another method, using a subset of samples with 5% lowest ratios and discarding the 3 lowest... I don't have the exact reference right now, but please add.

**Author's Response:** Suggestion taken. The reference (Pio et al., 2011) is now added in the main text.

> Lines 74-78
> "Combinations of the fixed percentile and the minimum $(OC/EC)_{pri}$ approaches were also used in order to accommodate different sample sizes available. For example, Pio et al. (2011) suggested using the lowest 5% subset to obtain the $(OC/EC)_{pri}$, and if the sample size of 5% subset is less than three, the lowest three data points are used to determine $(OC/EC)_{pri}$."

**Reference**
Pio, C., Cerqueira, M., Harrison, R. M., Nunes, T., Mirante, F., Alves, C., Oliveira, C., de la Campa, A. S., Artinano, B., and Matos, M.: OC/EC ratio observations in Europe: Re-thinking the approach for apportionment between primary and secondary organic carbon, Atmos Environ, 45, 6121-6132, DOI 10.1016/j.atmosenv.2011.08.045, 2011.

- line 90: any reason why the Millet method was overlooked?

**Author's Response:** One reason is that Millet's original paper focused on VOCs, and the MRS approach was used to calculate primary ratio of VOCs/EC to differentiate primary and secondary VOCs. A second reason we believe is a lack of evaluation work for this method. As a result, the approach initially proposed by Millet et al did not draw much attention from the OC/EC measurement community.

- line 211: please elaborate on why the OCEC10% method provides worse results

**Author's Response:** Based on the observational data we have, the ambient conditions most likely falls into the scenario between scenario A and B (Figure 3). As such, $OC/EC_{10\%}$ is further away from the true $OC/EC_{pri}$ than $OC/EC_{min}$, resulting larger bias.

- line 226: I don't understand the different behavior of the $OCEC_{10\%}$ amend the $OCEC_{min}$ methods, given that they are both subsets of the total dataset with specific characteristics of representing 1% and 10%. Why is their behavior different?

**Author's Response:** Change of $f_{SOC}$ not only changes the position of OC/EC distribution relative to $OC/EC_{pri}$ distribution, but can also alter the width of OC/EC distribution. Because the subset methods rely on percentile of OC/EC, once the OC/EC distribution is widened, the relative position between $OC/EC_{min}$ and $OC/EC_{10\%}$ is also changed and this results in a non-linear response in SOC differences,

- section uncertainty: with some analytical methods (e.g., TOT) the uncertainty is mostly constant (0,1-0,2 micrograms/cm2), please discuss how this would affect the results in this section.

**Author's Response:** Under the scenario of constant absolute uncertainty, the performance of MRS (Figure R1, 0.2 µg m$^{-3}$) is similar to that assuming a fix proportional measurement uncertainty (Figure R2, 10% measurement uncertainty). Both Figures R1 and R2 are included in the revised main text as Figure 8.

[Figure]

Figure R1. SOC estimation bias as a function of sample size, assuming fixed absolute measurement uncertainty for OC and EC (0.2 µgC m$^{-3}$). For each sample size, 500 repeat runs were conducted. The circles represent mean of 500 repeat runs, the whiskers represent one standard deviation. Parameters used for testing: Repeat runs = 500; N = 20~8000; EC = 8±4 µgC m$^{-3}$; $(OC/EC)_{pri}$ = 0.5; POC = 4 ±2 µgC m$^{-3}$, $f_{SOC}$ =40%, and SOC = 2.67±1.33 µgC m$^{-3}$.

[Figure]

Figure R2. SOC estimation bias as a function of sample size, assuming a fixed relative measurement uncertainty of 10% for OC and EC. For each sample size, 500 repeat runs were conducted. The open circle represents the mean of 500 repeat runs, and the whisker represents one standard deviation. Parameters used for testing: Repeat runs = 500; N = 8000; EC = 8±4 µgC m$^{-3}$; $(OC/EC)_{pri}$ = 0.5; POC = 4 ±2 µgC m$^{-3}$, $f_{SOC}$ =40%, and SOC = 2.67±1.33 µgC m$^{-3}$.

- line 317, please clarify what the authors mean by "the irrelevance of EC and SOC", it is unclear to me

**Author's Response:** We now rephrased as "the independence of EC and SOC", by which we mean that SOC and EC come from uncorrelated sources.

**Anonymous Referee #2**

**Generally Comments**

Typically the EC tracer method, when used in estimating the secondary organic carbon (SOC), relies on three conditions– 1) the relatively constant $(OC/EC)_{pri}$ over the period of study; 2) the random nature of SOC formation relative to EC; and 3) a subset of dataset without significant SOC contributions. The $OC/EC_{10\%}$ or $OC/EC_{min}$ essentially utilize the subset in Condition #3 to derive the $(OC/EC)_{pri}$ if it does have an unique value. Any deviations from the conditions as well as measurement uncertainties will lead to bias in determining $(OC/EC)_{pri}$. In some environments where SOC dominates, the third condition is generally impossible to be met. This study, through an extensive test, shows that the third condition is not necessary in calculating $(OC/EC)_{pri}$, if an algorithm, i.e., minimum $R^2$ (MRS), is used looking for $(OC/EC)_{pri}$ that yields SOC least correlated with EC. Without further examinations, the reviewer thinks that MRS is probably mathematically rigorous for any datasets satisfying the first two conditions and, additionally, with sufficient size and accuracy. It can perform better than $OC/EC_{10\%}$ or $OC/EC_{min}$ most of the time because Condition 3 is fortuitous, as described by the authors.

While the reviewer agrees that MRS should be used instead of $OC/EC_{10\%}$ or $OC/EC_{min}$ in calculating SOC, particularly for a large dataset which can support meaningful correlation analysis, MRS does not solve fundamental problems in the EC tracer method. The $(OC/EC)_{pri}$ is by no means constant, as it varies with source contributions from day to day and season to season. SOC is likely correlated with EC because in urban areas many SOC precursors originate from the same combustion sources as EC. This paper demonstrates that when Conditions 1 and 2 are in doubt, MRS produces erroneous results. MRS results are also sensitive to measurement uncertainty that impacts the correlation coefficients. These limitations, however, are not emphasized adequately in the abstract, which sounds almost like MRS has tackled all these issues. These issues, still, can only be solved by using multivariate or chemical mass balance analysis with additional markers.

**Author's Response:** Thanks for the very insightful comments. We agree that $(OC/EC)_{pri}$ varied from day to day and season to season in reality and this limitation is intrinsic in the EC tracer method regardless different approaches in implementing the EC tracer method, unless it is applied in a time frame small enough that variations of $(OC/EC)_{pri}$ is almost negligible. Limits posed by the nature of ambient ECOC data are inherent to the EC tracer method and common to all the variants of the EC tracer method. This study focuses on evaluating different $(OC/EC)_{pri}$ determination approaches within the EC tracer method, with the aim to identify the best approach in applying the EC tracer method. We have revised the wording in the abstract and in main text to emphasize the limitations of the EC tracer method and the MRS approach. Please see below the specific revisions in our point-by-point response to reviewers' comments.

**Specific Comments**

Abstract: Please describe the assumptions of MRS, datasets that are suitable for MRS analysis, and potential errors while in the same time shortening the abstract. Just saying MRS is better than $OC/EC_{10\%}$ or $OC/EC_{min}$ is not meaningful because all the three could be very wrong in some cases.

**Author's Response:** We have made the following revisions in the abstract to clearly state the assumptions of MRS.

Line 28:
"The hypothetical $(OC/EC)_{pri}$ that generates the minimum $R^2(SOC,EC)$ then represents the actual $(OC/EC)_{pri}$ ratio if variations of EC and SOC are independent and $(OC/EC)_{pri}$ is relatively constant in the study period."

Line 38-41:

"…MRS provides an unbiased SOC estimation when measurement uncertainty is small. MRS results are sensitive to the magnitude of measurement uncertainty but the bias would not exceed 23% if the uncertainty is controlled within 20%."

We also shortened slightly the abstract by condensing a few sentences and removing the following sentence (this background information is spelled out in the introduction section).

Line 97-102: While using simulated data is insightful, it offers no proof. The authors may explore if there is a true "proof" from mathematical or statistical derivations that MRS will yield true $(OC/EC)_{pri}$ if SOC is indeed random and the dataset is big enough. This may also answer the question- how big is big? MRS does not seem suitable for a dataset with only dozens of points.

**Author's Response:**
We agree that the simulated data alone does not offer proof, as there is no guarantee that the simulated data capture all the essential features of real-world data. In response to this comment, we conducted a series of sensitivity tests to evaluate the SOC estimation dependency on sample size, which was varied from 20 to 8000. For each sample size, 500 repeat runs were tested, assuming a single value $OC/EC_{pri}$ with a measurement uncertainty of 10%. The results are in Fig. R2, showing the average and the standard deviation for each sample size. The standard variation of SOC bias by MRS decreases with increased sample size while the mean of SOC bias remains a constant small value (2%). The standard variation of SOC bias is $\sim \pm 30\%$ at the lowest tested sample size ($n = 20$), and decreases to less than 15% at $n = 60$ (the sample size of one-year sampling from an every-six-day sampling program) and to less than 10% at $n = 200$. Other scenarios considering $OC/EC_{pri}$ with a distribution and different $f_{SOC}$ are discussed in SI. Figure R2 will be included in the main text.

A new section (as shown below) is added to the manuscript to address the sample size question.

[Figure]

Figure R2. SOC estimation bias as a function of sample size, assuming a fixed relative measurement uncertainty of 10% for OC and EC. For each sample size, 500 repeat runs were conducted. The open circle represents the mean of 500 repeat runs, and the whisker represents one standard deviation. Parameters used for testing: Repeat runs = 500; N = 8000; EC = 8±4 µgC m$^{-3}$; $(OC/EC)_{pri}$ = 0.5; POC = 1 ±0.5 µgC m$^{-3}$, $f_{SOC}$ =40%, and SOC = 0.67±0.34 µgC m$^{-3}$.

Lines 331-356:

"## 2.4 Impact of sample size

MRS rely on correlations of input variables and it is expected that MRS performance is sensitive to the sample size of input dataset. This section examines the sensitivity on sample size by the three $(OC/EC)_{pri}$ representations and aims to provide suggestions for an appropriate sample size when applying MRS on ambient OCEC data. Sample sizes ranging from 20 ~ 8000 are tested and for each sample size 500 repeat runs are conducted to obtain statistically significant results. Both Case A (i.e., a constant relative uncertainty of 10%) and Case B (i.e., a constant absolute uncertainty of ±0.2 μgC m$^{-3}$ for both OC and EC) are considered. The measurement uncertainties in case B are generated separately by MT following a uniform distribution within the range of ±0.2 μgC m$^{-3}$. The measurement uncertainties of POC and SOC are then back-calculated following the uncertainty propagation formula (Harris, 2010) and assuming the ratio of $\varepsilon_{POC}/\varepsilon_{SOC}$ is the same as POC/SOC ratio (controlled by $f_{SOC}$).

The mean SOC bias by MRS is very small (<3%) for all sample sizes while the standard deviation of SOC bias decreases with sample size (Figure 8). The standard deviation of SOC bias is ~±30% at the lowest test sample size ($n$ = 20), and decreases to less than ±15% at $n$ = 60 (the sample size of one-year sampling from an every-six-day sampling program) and to less than ±10% at $n$ = 200. Similar patterns are observed between Case A (Figure 8a) and Case B (Figure 8b) for MRS and $OC/EC_{10\%}$. For $OC/EC_{min}$, a larger bias is observed in Case B than Case A for all sample sizes, as SOC bias by $OC/EC_{min}$ is more sensitive to measurement uncertainty in the range of 0~10% as shown in Figure 7b. The standard deviation of SOC bias by $OC/EC_{min}$ and $OC/EC_{10\%}$ both decreases with sample size as shown in Figure 8. The mean SOC bias of $OC/EC_{min}$ decrease with increased sample size while $OC/EC_{10\%}$ is insensitive to sample size. The sample size dependency of all three $(OC/EC)_{pri}$ representations is not sensitive to $f_{SOC}$ as shown in Figure S16. Other scenarios considering $(OC/EC)_{pri}$ with a distribution and different $f_{SOC}$ are discussed in SI."

Line 116-118: How good are the K-S statistics? In other words, how well did the pseudorandom number generator reproduce the statistics in the original dataset?

**Author's Response:** The K-S statistics for ambient measured data are shown in Figures S1-S4 (This information is now also mentioned in the main text). In Igor Pro's Kolmogorov–Smirnov test, D represents the K-S statistic, C represent critical value. If D<C, the samples follow the corresponding distribution (e.g., normal or log-normal distribution). The majority of the data can pass the K-S test for log-normal distribution and some exhibit a bimodal distribution. For the performance of the MT pseudorandom number generator, we conduct a series of K-S tests on the generated data for 5000 runs, which show 94.4% data having D small than C (Fig. R3). Hence, we believe the pseudorandom number generator could produce the data following preset characteristics. Figure R3 is added to the SI and referred to in the main text.

[Figure]

Figure R3 Performance of the MT pseudorandom number generator evaluated by K-S test. The histogram in grey represents D statistic value in K-S test and the red dashed-line represents C. The dash line in green represents cumulative distribution of D. Data with D<C, i.e., data that strictly follow the log-normal distribution, account for 94.4% in 5000 runs.

The below text is added to the manuscript to describe whether the pseudorandom number generate reproduce the statistics in the original dataset.

Lines 142-145:
"For the verification of the log-normality of MT generated data, a series of K-S tests on the generated data for 5000 runs are conducted. As shown in Figure S6, 94.4% of runs pass the K-S test. Hence the performance of MT can satisfy the log-normal distributed data generation requirement in this study".

Line 126: Eqs. (4)-(5) do not work for all datasets. They are probably asymptotes when datasets are large enough in size.

**Author's Response:** We agree that they do not necessarily work for all datasets. The reason for translating mean and standard deviations into $\mu$ and $\sigma$ is that the MT pseudorandom number generator in Igor Pro only accepts $\mu$ and $\sigma$ as input parameters, while mean and standard deviations are the parameters that can be obtained from ambient measurements.

Line 136: Mention here that the case with combustion-related SOC is discussed later.

**Author's Response:** Suggestion taken. The text below is included in the revised manuscript:

Line 140:
"The case with combustion-related SOC is briefly discussed in section 3."

Line 151-152: The results of log-normally distributed $(OE/CC)_{pri}$ should be summarized in the text if possible.

**Author's Response:** Suggestion taken. The below text is added to the section 2.2.1:

Lines 219-225:
"For the representation of $(OC/EC)_{pri}$ in the simulated data as lognormally distributed data, analysis is also performed to evaluate SOC estimation bias as a function of $RSD_{EC}$, $RSD_{SOC}$, and $f_{SOC}$. Table

S2 summarizes the results obtained with adopting most probable ambient conditions (i.e., $RSD_{EC}$: 50-100%, $f_{SOC}$: 40-60%). SOC bias by MRS is within 4% when measurement uncertainty is ignored. In comparison, SOC bias by $OC/EC_{min}$ is more sensitive to assumption of log-normally distributed $(OC/EC)_{pri}$ than single value $(OC/EC)_{pri}$, including the dependency on $RSD_{EC}$ and $RSD_{SOC}$ with varied $f_{SOC.}$."

Line 220-222: It is not clear if $f_{EC1}$ was varied from sample to sample in a single test or only varied from test to test. If the former, how could you make sure EC1 and EC2 are highly correlated?

**Author's Response:**   $f_{EC1}$ was varied from test to test. The text is now clarified as below:

Lines 229-231:
"By varying $f_{EC1}$ (proportion of source 1 EC to total EC) from test to test, the effect of different mixing ratios of the two sources can be examined."

Line 284-286: Since POC and SOC are not directly measured, what is the meaning to simulate their measurement uncertainty?

**Author's Response:**   Once OC and EC data are considered to have measurement uncertainty, the derived quantities POC and SOC (using Eq (1) and Eq (2)) consequently also have associated uncertainty, which can be calculated following uncertainty propagation principle. For the evaluation of SOC estimation, SOC calculated from the EC tracer method needs to be compared with "true SOC plus associated uncertainty". That's the reason why we calculated the uncertainties of POC and SOC

Line 384: How were the six subsets selected?

**Author's Response:** With a given one-year data set, there are six possible extractions of daily data sets corresponding to the assumed every-six-day sampling schedule, i.e., set 1:{Day 1, 7, 13,..}, set 2: {Day 2, 8, 14,..}, set 3: {Day 3, 9, 15,..}, etc. The text below is added to clarify this point:

Lines 368-371:
"The one-year data yields six subsets of daily samples, corresponding to six possible schedules of sampling days with the every-six-day sampling frequency. The MRS calculation produces the $OC/EC_{pri}$ in the range of $2.37 - 2.75...$"

Line 360-362: Emphasize that this only happens when measurement uncertainties are small.

**Author's Response:** Suggestion taken. This sentence is revised as below:

Lines 408-413:
"In the scenarios of a single primary source and two well-correlated primary combustion sources, SOC estimates by MRS are unbiased while $OC/EC_{min}$ and $OC/EC_{10\%}$ consistently underestimate SOC when measurement uncertainty is neglected. When measurement uncertainty is considered, all three approaches produce biased SOC estimates, with MRS producing the smallest bias. The bias by MRS is less than 25% when measurement uncertainty is within 20% and $f_{SOC}$ is not lower than 20%."

[revised manuscript text omitted]

The Mersenne twister (MT) (Matsumoto and Nishimura, 1998), a pseudorandom number generator, is used in data generation. MT is provided as a function in Igor Pro. The system clock is utilized as the initial condition for generation of pseudorandom numbers. The data generated by MT has a very long period of $2^{19937} - 1$, permitting large data size and ensuring that pseudorandom numbers are statistically independent between each data generation. The latter feature ensures the independent relationship between EC and non-combustion related SOC data. The case with combustion-related SOC is briefly discussed in section 3. MT also allows assigning a log-normal distribution during pseudorandom number generation to constrain the data. For the verification of the log-normality of MT generated data, a series of K-S tests on the generated data for 5000 runs are conducted. As shown in Figure S6, 94.4% of runs pass the K-S test. Hence the performance of MT can satisfy the log-normal distributed data generation requirement in this study. 
[revised manuscript text omitted]

For the representation of $(OC/EC)_{pri}$ in the simulated data as lognormally distributed data, analysis is also performed to evaluate SOC estimation bias as a function of $RSD_{EC}$, $RSD_{SOC}$, and $f_{SOC}$. Table S2 summarizes the results obtained with adopting most probable ambient conditions (i.e., $RSD_{EC}$: 50-100%, $f_{SOC}$: 40-60%). SOC bias by MRS is within 4% when measurement uncertainty is ignored. In comparison, SOC bias by $OC/EC_{min}$ is more sensitive to assumption of log-normally distributed $(OC/EC)_{pri}$ than single value $(OC/EC)_{pri}$, including the dependency on $RSD_{EC}$ and $RSD_{SOC}$ with varied $f_{SOC}$.

**2.2.2 Scenarios assuming two primary sources**

In the real atmosphere, multiple combustion sources impacting a site is normal. We next evaluate the performance of the MRS method in scenarios of two primary sources and arbitrarily dictate that the $(OC/EC)_{pri}$ of source 1 is lower than source 2. By varying $f_{EC1}$

(proportion of source 1 EC to total EC) from test to test, the effect of different mixing ratios of the two sources can be examined. Common configurations in S2 and S3 include:

[revised manuscript text omitted]

**2.4 Impact of sample size**

MRS relies on correlations of input variables and it is expected that MRS performance is sensitive to the sample size of input dataset. This section examines the sensitivity on sample size by the three $(OC/EC)_{pri}$ representations and aims to provide suggestions for an appropriate sample size when applying MRS on ambient OCEC data. Sample sizes ranging from $20 \sim 8000$ are tested and for each sample size 500 repeat runs are conducted to obtain statistically significant results. Both Case A (i.e., a constant relative uncertainty of 10%) and Case B (i.e., a constant absolute uncertainty of $\pm 0.2$ $\mu gC$ $m^{-3}$ for both OC and EC) are considered. The measurement uncertainties in case B are generated separately by MT following a uniform distribution within the range of $\pm 0.2$ $\mu gC$ $m^{-3}$. The measurement uncertainties of POC and SOC are then back-calculated following the uncertainty propagation formula (Harris, 2010) and assuming the ratio of $\varepsilon_{POC}$ /$\varepsilon_{SOC}$ is the same as POC/SOC ratio (controlled by $f_{SOC}$).

The mean SOC bias by MRS is very small (<3%) for all sample sizes while the standard deviation of SOC bias decreases with sample size (Figure 8). The standard deviation of SOC bias is $\sim\pm 30\%$ at the lowest test sample size ($n = 20$), and decreases to less than $\pm 15\%$ at $n = 60$ (the sample size of one-year sampling from an every-six-day sampling program) and to less than $\pm 10\%$ at $n = 200$. Similar patterns are observed between Case A (Figure 8a) and Case B (Figure 8b) for MRS and $OC/EC_{10\%}$. For $OC/EC_{min}$, a larger bias is observed in Case B than Case A for all sample sizes, as SOC bias by $OC/EC_{min}$ is more sensitive to measurement uncertainty in the range of 0~10% as shown in Figure 7b. The standard deviation of SOC bias by $OC/EC_{min}$ and $OC/EC_{10\%}$ both decreases with sample size as shown in Figure 8. The mean SOC bias of $OC/EC_{min}$ decrease with increased sample size while $OC/EC_{10\%}$ is insensitive to sample size. The sample size dependency of all three $(OC/EC)_{pri}$ representations is not sensitive to $f_{SOC}$ as shown in Figure S16. Other scenarios considering $(OC/EC)_{pri}$ with a distribution and different $f_{SOC}$ are discussed in SI.

**2.5 Impact of sampling time resolution**

[revised manuscript text omitted]

**Figure 7.** Bias of SOC determination as a function of relative measurement uncertainty ($\gamma_{unc}$) and SOC/OC ratio ($f_{SOC}$) by different approaches of estimating $(OC/EC)_{pri}$, including ratio of averages (ROA), minimum R square (MRS), $OC/EC_{10\%}$, and $OC/EC_{min}$. Fixed input parameters: $n$=8000, EC = $2\pm1$ μgm$^{-3}$, $(OC/EC)_{pri}$ = 0.5. Variable input parameters: (a) $f_{SOC}$ =20%, SOC = $0.25\pm0.13$ μgC m$^{-3}$, (b) $f_{SOC}$ =40%, SOC = $0.67\pm0.33$ μgC m$^{-3}$, (c) $f_{SOC}$ =60%, SOC = $1.5\pm0.75$ μgC m$^{-3}$, and (d) $f_{SOC}$ =80%, SOC = $4\pm2$ μgC m$^{-3}$

[Figure]

**Figure 8.**  SOC estimation bias as a function of sample size by different approaches of estimating $(OC/EC)_{pri}$, including minimum R square (MRS), $OC/EC_{10\%}$, and $OC/EC_{min}$, (a) assuming a fixed relative measurement uncertainty of 10% for OC and EC; (b) assuming a fixed absolute measurement uncertainty for OC and EC (0.2 μg m$^{-3}$). For each sample size, 500 repeat runs were conducted. The circles represent mean of 500 repeat runs, the whiskers represent one standard deviation. Parameters used for testing: Repeat runs = 500, $n$ = 20~8000, EC = 8±4 μgC m$^{-3}$, $(OC/EC)_{pri}$ = 0.5, POC = 4±2 μgC m$^{-3}$, $f_{SOC}$ =40%, and SOC = 2.67±1.33 μgC m$^{-3}$.

[Figure]

**Figure 9.** OC/EC distributions assuming different average intervals from 2 to 24 h and the corresponding MRS-derived OC/EC$_{pri}$. The bottom x-axis represents averaging interval (e.g. 1 h is the original data time resolution, 2 h referring average 1-h data into 2-h interval data, etc). The top x-axis represents the number of data point corresponding to the respective data averaging interval. Distributions of OC/EC ratio at various averaging intervals are shown as box plots (Empty circles: average, the line inside the box: median, the box boundaries: 75$^{th}$ and the 25$^{th}$ percentile, and the whiskers: 95$^{th}$ and 5$^{th}$ percentile). The red dots represent calculated (OC/EC)$_{pri}$ by MRS.